# Uniaxial Cyclic Cell Stretching Device for Accelerating Cellular Studies

**DOI:** 10.3390/mi14081537

**Published:** 2023-07-31

**Authors:** Sharda Yadav, Pradip Singha, Nhat-Khuong Nguyen, Chin Hong Ooi, Navid Kashaninejad, Nam-Trung Nguyen

**Affiliations:** Queensland Micro- and Nanotechnology Centre (QMNC), Griffith University, Nathan, QLD 4111, Australia; s.yadav@griffith.edu.au (S.Y.); p.singha@griffith.edu.au (P.S.); nhatkhuong.nguyen@griffith.edu.au (N.-K.N.); c.ooi@griffith.edu.au (C.H.O.); n.kashaninejad@griffith.edu.au (N.K.)

**Keywords:** mechanobiology, cell stretching, biomedical device, extracellular matrix

## Abstract

Cellular response to mechanical stimuli is a crucial factor for maintaining cell homeostasis. The interaction between the extracellular matrix and mechanical stress plays a significant role in organizing the cytoskeleton and aligning cells. Tools that apply mechanical forces to cells and tissues, as well as those capable of measuring the mechanical properties of biological cells, have greatly contributed to our understanding of fundamental mechanobiology. These tools have been extensively employed to unveil the substantial influence of mechanical cues on the development and progression of various diseases. In this report, we present an economical and high-performance uniaxial cell stretching device. This paper reports the detailed operation concept of the device, experimental design, and characterization. The device was tested with MDA-MB-231 breast cancer cells. The experimental results agree well with previously documented morphological changes resulting from stretching forces on cancer cells. Remarkably, our new device demonstrates comparable cellular changes within 30 min compared with the previous 2 h stretching duration. This third-generation device significantly improved the stretching capabilities compared with its previous counterparts, resulting in a remarkable reduction in stretching time and a substantial increase in overall efficiency. Moreover, the device design incorporates an open-source software interface, facilitating convenient parameter adjustments such as strain, stretching speed, frequency, and duration. Its versatility enables seamless integration with various optical microscopes, thereby yielding novel insights into the realm of mechanobiology.

## 1. Introduction

Mechanotransduction is the process that converts mechanical stimuli to biochemical activities in cells [1,2]. Notably, mechanosensitivity is a fundamental feature shared by all living organisms, underscoring its ubiquity and significance [2,3,4]. The indispensable role of mechanotransduction in cell development is manifested through mechanical interactions among neighboring cells and the extracellular matrix (ECM) [5,6,7]. For instance, endothelial cells endure constant mechanical stresses, exemplifying the impact of mechanotransduction on cellular behavior [8,9]. Similarly, myocytes have been reported to reorientate and elongate along the stretch axis [10]. Furthermore, mechanotransduction has emerged as a crucial factor in the etiology of certain pathological conditions, including cardiovascular diseases [11] and specific types of cancer [12], thereby emphasizing its clinical relevance [13,14].

Most in vivo cells are subjected to a range of mechanical forces, such as tensile, compressive, and shear forces, that act externally through their ECM [15,16]. At the cellular level, these forces significantly affect the basic functions of cells, such as gene expression, morphology, proliferation, differentiation, and spreading [17,18]. At the tissue and organ levels, these mechanical forces play important roles in tissue function, organoid formation, immune response, wound healing, embryonic development, and cancer formation and metastasis [19]. Therefore, a fundamental understanding of how cells can sense these mechanical forces and convert them into biochemical responses, called mechanotransduction, is of paramount importance for both physiological and pathophysiological studies. Accordingly, an ideal in vitro cell culture platforms need to recapitulate both biological and mechanical environments of cells and tissues in vivo [20].

To design such an in vitro cell culture platform for mechanical stimulations, three crucial factors need to be considered: the direction, magnitude, and duration of applied forces [21]. Furthermore, surface properties, including the cell-surface receptor affinities, mechanical stiffness, biocompatibility, degradability, and viscoelasticity of a biomaterial used as a substrate to support cells in these mechanical stimulation devices, need to be carefully considered based on the in vivo conditions [19]. For instance, tendon cells anchored to collagen fibers can return to their original size and shape if a small magnitude of tensile stress is applied over an extended period [22,23]. However, these cells may deform permanently or even rupture if they are stretched largely and rapidly. Other cells, such as those that exist in blood vessels (e.g., endothelial cells), cardiovascular system (e.g., cardiomyocytes and vascular smooth muscle cells), cartilage (e.g., chondrocytes), and bones (e.g., osteocytes), are subject to different ranges of combined tensile, compressive, and shear forces [24]. These forces encompass a wide range of amplitudes and frequencies, acting upon living cells either in a static or cyclic manner [25].

One of the most popular techniques to study mechanotransduction is characterizing cell development using a cell stretcher. A cell stretcher is a device that can induce static or cyclical strain on the cells adhered to a flexible substrate. Researchers can accurately control the amount of strain to study the desired responses. Sophisticated atomic force microscopy (AFM) combined with optical imaging can be used to precisely quantify the mechanical properties of the deformed cells at the single cell level [26]. Commercial cell stretching platforms are available in the market, albeit at a prohibitively high cost of over ten thousand dollars. Furthermore, it is important to note that existing devices for cell stretching are often characterized by low throughput and an inability to generate a perfectly linear stress−strain profile [27]. Although several alternative designs for cell stretchers have been proposed [28,29,30,31,32,33,34], they still entail trade-offs in terms of build simplicity, user-friendliness, scalability, cost, and performance.

A cell stretcher typically consists of two primary components. First and foremost is the flexible container responsible for holding the cell culture. This container is typically constructed using a biocompatible material, considering its direct physical interaction with the cells. Polydimethylsiloxane (PDMS) is widely employed as one of the most used materials due to its biocompatibility, ease of casting, transparency, wide availability, and adjustable hardness. The second component is the auxiliary platform that drives the container. This platform provides an accurate uniaxial or biaxial motion that stretches the container. Various actuation schemes have been employed, including electromagnetic [35], piezoelectric [36], mechanical [30], pneumatic [37,38], and optical actuators [25,27]. Mechanical actuation schemes usually involve a motorized driver. Therefore, the following discussion broadly categorizes cell stretchers into motorized and non-motorized ones.

Non-motorized cell stretchers involve an additional layer of complexity due to indirect stretching. Specifically, the elongation of the container depends on a secondary parameter determined by the actuation scheme. An example of such a stretcher is an electromagnet-based system that generates an electromagnetic force. This force is affected by factors such as magnetic permeability, magnetic flux density, magnet geometry, and the distance between the magnets. These parameters play a significant role in determining the magnitude and characteristics of the stretching force applied by the electromagnet [35]. A pneumatic stretcher is affected by variations in the membrane thickness, input pressure, ambient temperature, and even humidity [37,38]. These examples highlight the rigorous, yet essential calibration processes required to achieve reproducibility in cell stretching experiments. While certain commercial pneumatic platforms may advertise a small system footprint, it is important to note that this footprint does not encompass the compressed air supply. In practical terms, the infrastructure requirements are high as the system can only operate in locations equipped with existing compressed air lines or require an air pump, often with a large physical footprint. Additionally, the requirement for compressed air or vacuum necessitates the use of connecting tubes, which result in cumbersome handling. Pneumatic platforms employing soft membranes may also be prone to leaks during extended cyclic operations, posing challenges to their reliability and performance.

Conversely, motorized cell stretchers, especially those using high-precision stepper motors, can achieve large and consistent elongation with a relatively simple setup [39]. In addition, motorized actuators are generally highly stable throughout time, making them suitable for both static and cyclic stretching. Characterization of the elongation is straightforward as the user input elongation is directly related to the output elongation without involving any secondary parameter. This task occurs in the motor and controller, which enables highly precise and programmable motions. Generally, motorized stretchers tend to have the least requirements for infrastructure. Most motorized stretchers only require ubiquitous electrical input. Atcha et al. reported a low-cost motorized cell stretcher device [40]. By utilizing a programmable servomotor, gear, and gear rack system, the team devised a versatile cell stretching platform to uniaxially strain macrophages and cardiomyocytes in a cyclic manner. A recent motorized cell stretcher built with LEGO parts by Boulter et al. is even more impressive as it runs on a small battery pack, effectively removing any external connections during operation [29]. Such a portable setup is possible due to the availability of highly energy-efficient motor designs. 

Two of the most recent actuators are of particular interest. The first one is a platform developed by Al-Maslamani et al. [28]. The team used a low-cost linear actuator mounted on a 3D printed stage that can be controlled wirelessly. The relatively small footprint allows for fitting the platform inside an incubator and placing it on a microscope stage. The accuracy is acceptable at +/−0.3 mm for large strains, and there is an ample amount of force available up to 22 N.

On the other hand, Boulter et al. developed a novel cell stretching platform using strong and lightweight LEGO parts [29]. However, the small LEGO motor can only produce a limited force. Furthermore, different gear ratios must be used to achieve suitable stretching frequencies. This substantially increases the operational complexities of changing the gear ratio, which also affects the output force. Although the concept does not require any 3D printer to build, the assembly process is complex as it involves a long list of parts and instructions. In addition, the combined LEGO parts are expensive compared with a monolithic 3D-printed platform.

The present paper reports a novel uniaxial cell stretcher that leverages a high-precision motorized linear stage controlled by an Arduino microcontroller. Our cell stretcher offers the flexibility to be positioned on a microscope stage, enabling direct observation of the cell stretching process. Furthermore, it can be seamlessly integrated into an incubator, allowing for the implementation of the stretching process during cell culture procedures. A key advantage of our developed cell stretcher is its modular design, facilitating integration with various applications and enabling straightforward assembly. The primary objective of this study is to introduce a cost-effective and high-performance uniaxial cell stretching device, providing a detailed operational concept, experimental design, and characterization. The study focuses on testing the device with MDA-MB-231 breast cancer cells. The aim is to demonstrate cellular changes within a 30 min stretching duration compared with the previously required 2 h duration. This third-generation device significantly improved the stretching capabilities compared with its previous counterparts, resulting in a remarkable reduction in stretching time and a substantial increase in overall efficiency.

## 2. Materials and Methods

### 2.1. Cell Stretching Platform

A platform was designed to accommodate the motorized stage and the PDMS container (Figure 1a). The entire platform was constructed with polymethyl methacrylate (PMMA) plates. The geometries are simple with a straightforward fabrication process. However, the two stretching arms were specifically crafted using metal angle brackets, measuring 2 mm in thickness. Instead of 3D printing, we used a 2D design to build the frame. Each component was drawn and cut from 6 mm thick PMMA slabs using a CO_2_ laser cutter (Rayjet 300 from Trotec Laser, Loganholme, Australia). The slabs could also be sawn or milled, depending on the user preference. This enables a truly rapid prototyping process as the 2D drawing and cutting processes can be completed in minutes instead of hours, as required for 3D printing. The components were attached using adhesive, as well as bolts and nuts.

### 2.2. Motorized Stage and Controller

The linear stage is driven by a two-phase 1.8° JKM NEMA17 42 mm hybrid stepper motor. The motorized linear stage is rated for a load up to 80 N. The high output force substantially improves throughput by enabling running multiple stretching assays in parallel. The worm drive enables high-precision motion and repeatability at +/−0.01 mm. This stage is commercially available for a price tag of less than 60 USD. The motor is controlled using an Arduino (Uno) controller. Figure 1b, and Appendix A shows the operational scheme. Using open-source software, the user can easily change the strain, stretching speed, frequency, and duration.

### 2.3. PDMS Container

The PDMS container was cast using the same materials and methods from our previous electromagnetic cell stretcher [35], except that no permanent magnet was embedded for actuation. A thin PDMS membrane was attached to the bottom of the container using plasma treatment to form a watertight seal. Subsequently, the culture medium containing cells could be deposited on the membrane. When the stretching, culturing, and analysis were completed, we peeled off the membrane from the container and stored it for further analysis. The container could be disinfected for repeated use. This process substantially reduced the materials used for each run as it only consumed a thin piece of membrane instead of an entire block of PDMS. Moreover, curing a thin piece of a membrane was much faster than a bulky piece of PDMS.

### 2.4. Cell Culture, Maintenance, and Co-Culture Using the Cell Stretching Device

Human breast cancer cells (MDA-MB-231) were obtained from the American Type of Culture Collection (ATCC, Manassas, VA, USA). Dulbecco’s Modified Eagle Medium/Nutrient Mixture F-12 (DEME/F12), heat-inactivated fetal bovine serum (FBS), and penicillin/streptomycin were purchased from the Gibco-Thermo Fisher Scientific (Waltham, MA, USA). MDA-MB-231 cells were cultured in DEME/F-12, 10% FBS, and 1% penicillin/streptomycin in a humidified atmosphere at 37 °C, 5% CO_2_. The device was sterilized with 80% ethanol and washed three times with sterile 1X Hank’s balanced salt solution (HBSS). Ultraviolet (UV) irradiation was applied for 30 min, 400 µL of DMEM-F12 media was then added to the cell-stretching device and incubated for 1 h to further enhance biocompatibility. Once the culture reached 80% confluence, the cells were harvested from the flasks and counted using a hemocytometer. In total, we seeded 50,000 cells onto the PDMS membrane. The cells were placed inside the incubator for 24 h to optimize their adhesion and growth on the PDMS membrane. Subsequently, the cultured cells were washed (3×) with HBSS and topped up with 400 µL of the medium. The 5%, 10%, and 20% strains were applied, as explained in Section 3.

### 2.5. Operation of the Cell Stretching Device

Next, the PDMS container with fully grown cells was placed between the two arms of the stretching device and placed inside the incubator (Figure 1c). The distance between the two stretching arms was kept at 4 mm before all experiments to properly fit the PDMS container within them. We switched on the motorized-stage and ran the Arduino program with the set values for stretching. The actuating arms started to stretch the cells for several cycles. We controlled the frequency, strain, and number of cycles through the microcontroller. After completing the stretching process, we took the device along with the PDMS container out of the incubator. The PDMS container with the stretched cells was then removed for further analysis, and the device was disinfected before the next run.

### 2.6. Immunofluorescence Staining

The cells were fixed by treating them with 4% paraformaldehyde (PFA) for 15 min on the membrane. Subsequently, the cells were washed three times with PBS. Standard immunofluorescence staining was employed to examine the actin filaments and nuclei of the cells on the PDMS membrane before and after stretching. The fixed cells were stained with ActinGreen^TM^ 488 (Thermo Fisher Scientific) and NucBlue^TM^ ReadyProbe^TM^ reagents (Thermo Fisher Scientific) and incubated at room temperature for 30 min. Subsequently, the cells were washed three times with PBS. The membrane was then stored at 4 °C for subsequent imaging and analysis. The actin fibers and cell nuclei were observed using an inverted microscope (Nikon Eclipse Ti2; Nikon, Tokyo, Japan), as described in the following section.

### 2.7. Image Analysis of the Stretched Cells

We initially separated the PDMS membrane containing the stained cells by cutting, and detaching, and placedit onto a microscope slide to obtain cell images (Figure 1d). A fluorescence microscope (Nikon Eclipse Ti2) was utilized to capture images of the actin fibers and cell nuclei. Following image acquisition, post-processing was carried out using Image J 1.47v software (National Institutes of Health (NIH), Bethesda, MD, USA). Cell images were captured at three different locations. For analysis, a minimum of 100 cells from three distinct regions were examined across three biological repetitions. The outline of each cell was manually traced to measure its volume and perimeter. The aspect ratio, the ratio between the major and minor axes of a fitted ellipse, and the length as represented by the major axis were determined. Subsequent data analysis was conducted using GraphPad Prism 7 (GraphPad Prism Software Inc., San Diego, CA, USA).

### 2.8. Statistical Analysis

All experimental data were captured quantitatively, and statistical analysis was performed using the Statistical Package for Social Sciences for Windows (version 22.0, IBM SPSS Inc., New York, NY, USA). Independent *t*-test and ANOVA were performed for the analysis of continuous variables in categories. A significance level of the tests was taken at *p*  <  0.05.

## 3. Results

### 3.1. Performance of the Cell Stretcher

We carried out a strain analysis of the membrane in action to understand the performance of our device. A USB camera captured the deformation of the membrane. The strain calculated from the experiments was compared to that of the input into the microcontroller (Figure 2). Each data point in the figure represents the average value of three trials with standard deviation as an error bar. For a set input of 5% strain, the actual measured strain of the membrane was 4.18 ± 0.63%. According to this analysis, the cell stretcher provided an average homogeneous cyclic strain with an acceptable error of 10%.

On the other hand, we calculated the force required for stretching the PDMS container using Hooke’s law. Assuming a spring constant of 2.41 N/mm for the PDMS container [35], we found that only a force of 1.9 N was needed to achieve 10% strain. Hence, we believe that our stretching device with a maximum load of 80 N can stretch the PDMS container with a higher strain if needed. Moreover, this powerful stretcher can deform containers made with materials more rigid than PDMS. The incubator temperature was maintained at 37 °C, which is well within the operational range from −10 °C to 50 °C of our stretching device.

### 3.2. Parameter Optimization

The checkerboard method for stretching parameter optimization showed that 5% of strain applied at the frequency of 0.1 Hz demonstrated a difference in cellular parameters compared to that generated by the electromagnetic stretching device. Initially, we maintained a constant frequency of 0.1 Hz while varying the strain parameters, specifically 5%, 10%, and 20%. Following a 30 min stretching session, we carefully measured the cell length and cell area. Remarkably, we observed a significant increase in cell length as the strain rate was elevated. Additionally, a notable change in cell area was observed between the control group and the 5% strain group (Figure 3a,b). Subsequently, we optimized the frequency by fixing the strain rate at 5%. Figure 3c shows that the cell length increased as the frequency increased. However, Figure 4d highlights a significant difference between the control group and those under 0.1 Hz actuation, whereas no substantial changes were observed as the frequency further increased.

Similarly, we performed a frequency optimization at a fixed strain of 5%. Figure 3c evidently demonstrates that the length of the cells increased with higher frequencies. Furthermore, Figure 3d highlights significant changes between the control group and those under 0.1 Hz actuation. However, no noticeable changes were observed as the frequency was increased beyond that point. After carefully analyzing the experimental data and considering the observed trends, we ultimately focused on optimizing the specific parameter combination of applying a 5% strain at a frequency of 0.1 Hz. This particular configuration displayed significant effects on cellular parameters that need to be investigated.

### 3.3. Cellular Rearrangement

In order to gain a deeper understanding of the functionality of our device, we analyzed the rearrangement of cellular structures and changes in actin filaments when cells are subjected to stretching. Our initial observations regarding the cellular changes agreed with previous studies [41], validating the efficacy of the developed cell stretching platform. As anticipated, the reorganization of actin stress fibers appeared to facilitate dynamic cell adhesion and induce alterations in cell morphology. Notably, after 30 min of cell stretching, we observed a significant formation of cell clusters (Figure 4). The enhanced cellular connections and the clustering of cells illustrate how individual cells perceive and transmit physical forces to neighboring cells through the binding of adhesion molecules, thereby strengthening cell−cell cohesion. The proportion of clustering of the cells was calculated by subtracting the single cells from the total of 100 cells counted, which was used for calculating the cellular parameters. We noted that only 63% of cells were clustered with stretching for 10 min, 82% clustered with 20 min stretching, 86% clustered with 30 min stretching, and 65% clustered in the non-stretched control. Furthermore, we noticed that the cells responded to external stress transmission by remodeling their cytoskeletal architecture and reconstructing the actin stress fibers over time. Furthermore, in line with our expectations for non-stretched (control) conditions, no significant alterations were observed.

Additionally, we analyzed the morphological parameters to gain further insights (Figure 5). These parameters included cell area, perimeter-to-area ratio (cellular roundness), aspect ratio (cell elongation), and cell length. Using this device, we observed a distinct growth pattern. Specifically, the cell area began to increase only after 20 min of stretching, as expected, while the cell roundness decreased. Interestingly, the length of the cells showed a gradual increase after 20 min of stretching, reaching its maximum at the 30 min mark. However, no significant changes were observed in the aspect ratio.

It is noteworthy to mention that in previous studies conducted by Yadav et al. [41,42,43] and Kamble et al. [35], similar morphological changes were observed after a longer duration of 2 h using electromagnetically driven cell stretching devices. In contrast, the present uniaxial cell stretching platform allowed for achieving comparable changes within a significantly shorter duration of 30 min. These findings not only contributed to a better understanding of the functionality of the device, but also confirm its suitability for subsequent cell stretching experiments.

## 4. Conclusions

We successfully developed a cell stretching platform with a simple yet efficient method for exposing a cell culture to cyclic and static strains in a uniform manner. To evaluate the effectiveness of our platform, we conducted stretching assays with breast cancer cells. Our initial analysis revealed a strong correlation between the orientation of cells and the external mechanical cues. Notably, when subjected to cyclic stretching for a duration of 30 min, we observed cell aggregation, suggesting that the cells underwent a cytoskeletal reorganization to withstand the applied strain and to maintain the integrity of their ECM arrangement.

While extensive investigations with this device are ongoing, our initial findings have demonstrated its remarkable capability to induce similar cellular changes within the order of minutes compared with the hours typically needed by electromagnetically actuated devices. This accelerated response time is achieved under reproducible and repeatable conditions. The design of our platform allows users to easily adjust parameters such as strain, stretching speed, frequency, and duration through an open-source software interface. This enhanced efficiency in generating cellular responses will be particularly valuable for primary cells isolated from clinical samples.

Moreover, compared with our first-generation pneumatic device and second-generation electromagnetic device, this third-generation device holds the potential for multiplexing, enabling the processing of multiple samples simultaneously. The device is capable of simultaneous imaging during the stretch cycles providing capability of real-time monitoring of biomechanical activities by simply placing the device on the microscope stage. This capability is crucial for conducting clinical cohort studies, where large sample sizes are often necessary. It also enhances the efficiency and throughput of the cell stretching process, enabling researchers to handle and analyze multiple samples concurrently.

Additionally, it is worth highlighting that the cells can be harvested after stretching using conventional clinical tools, enabling subsequent standard biological analyses. This feature is of particular importance for clinical diagnostics and subsequent therapeutic screening.

## Figures and Tables

**Figure 1 micromachines-14-01537-f001:**
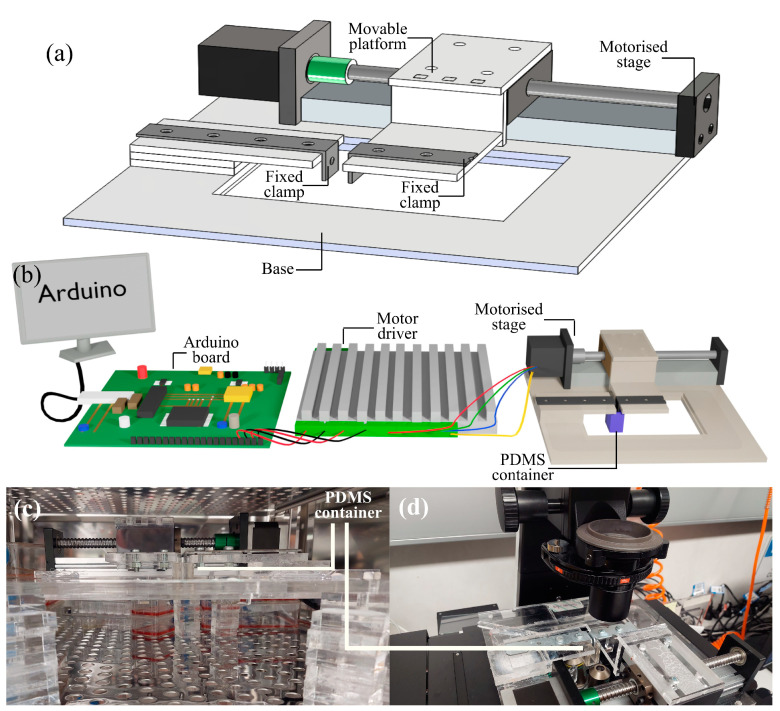
(**a**) Schematic illustration of the cell stretching device. (**b**) Complete setup of the cell stretching device connected to an Arduino controller. (**c**) The cell stretching device positioned inside a standard incubator. (**d**) The cell stretching device placed under a conventional microscope stage for observing the cell behavior.

**Figure 2 micromachines-14-01537-f002:**
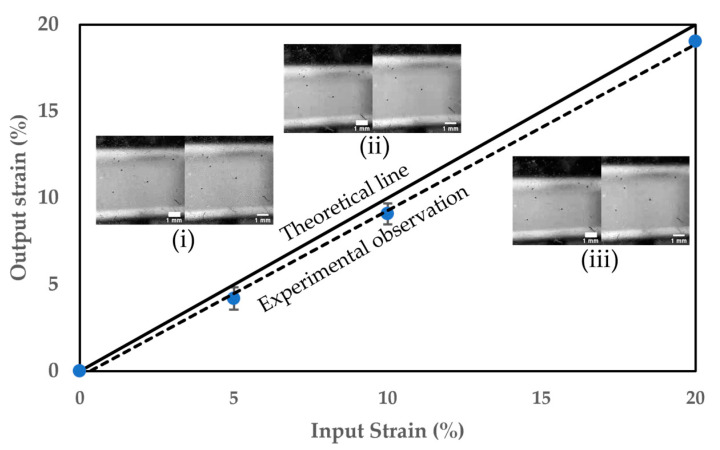
Device characterization: Set input strain vs. actual output strain. Inset images show the bottom view of the PDMS chamber before (left side) and after (right side) for (**i**) 5% strain, (**ii**) 10% strain, and (**iii**) 20% strain. The scale bars of the insets are 1 mm.

**Figure 3 micromachines-14-01537-f003:**
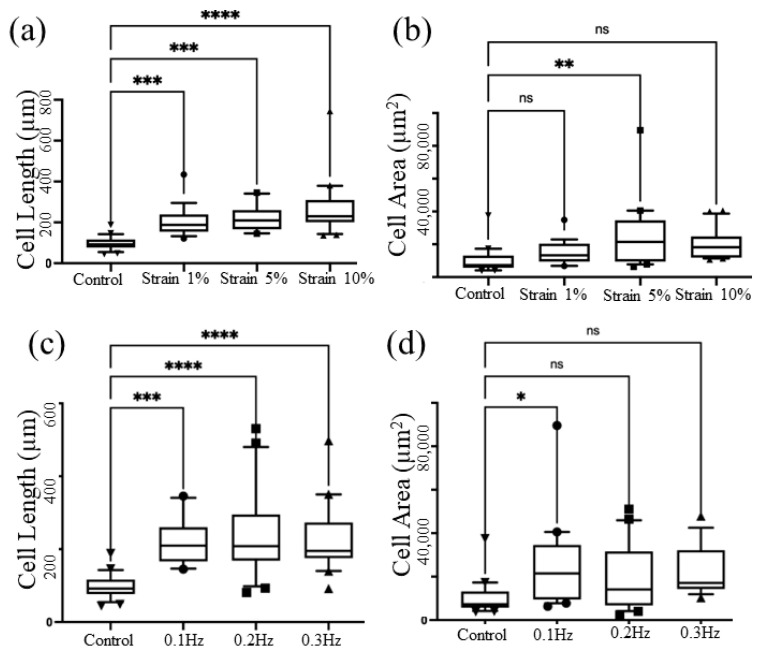
Parameter optimization: Measurement of the cell length (**a**) and cell area (**b**) by applying a frequency of 0.1 Hz while varying the strain parameters, specifically 1%, 5%, and 10%. Measurement of the cell length (**c**) and cell area (**d**) by applying 5% strain rate while varying the frequency, such as 0.1 Hz, 0.2 Hz, and 0.3 Hz. The asterisks denote statistically significant (*p*  <  0.05) differences as determined using multivariate analysis (ANOVA), * (*p* ≤ 0.05), ** (*p* ≤ 0.01), *** (*p* ≤ 0.001), **** (*p* ≤ 0.0001) and ns (*p* > 0.05).

**Figure 4 micromachines-14-01537-f004:**
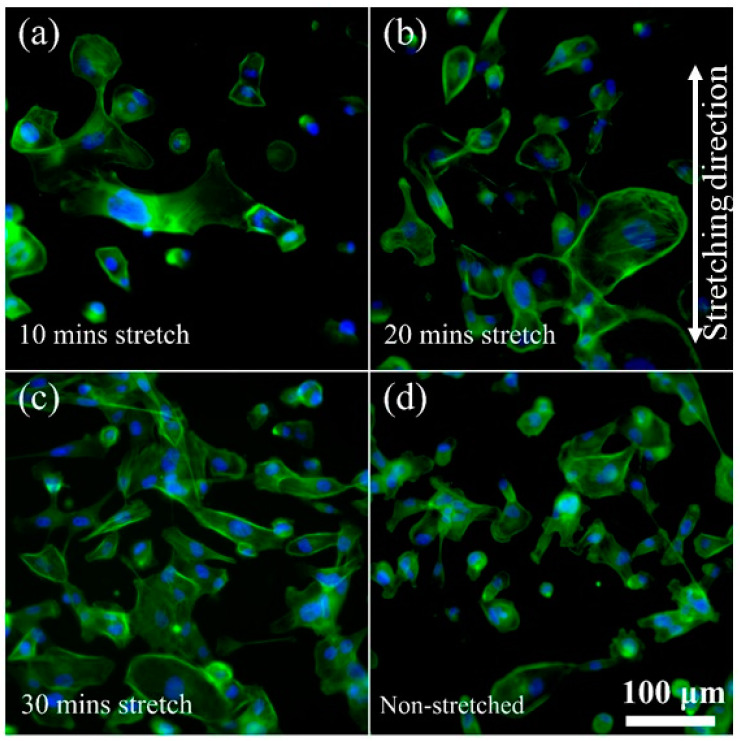
Representative fluorescence images of MDA-MB-231 cells showing cell morphology and distribution after stretching for (**a**) 10 min, (**b**) 20 min, and (**c**) 30 min compared to the (**d**) non-stretched cells. In the fluorescence images, actin is labelled with ActinGreen (green), and nuclei are labelled with Nucblue (blue). The cells shown are representative of the data (n = 3).

**Figure 5 micromachines-14-01537-f005:**
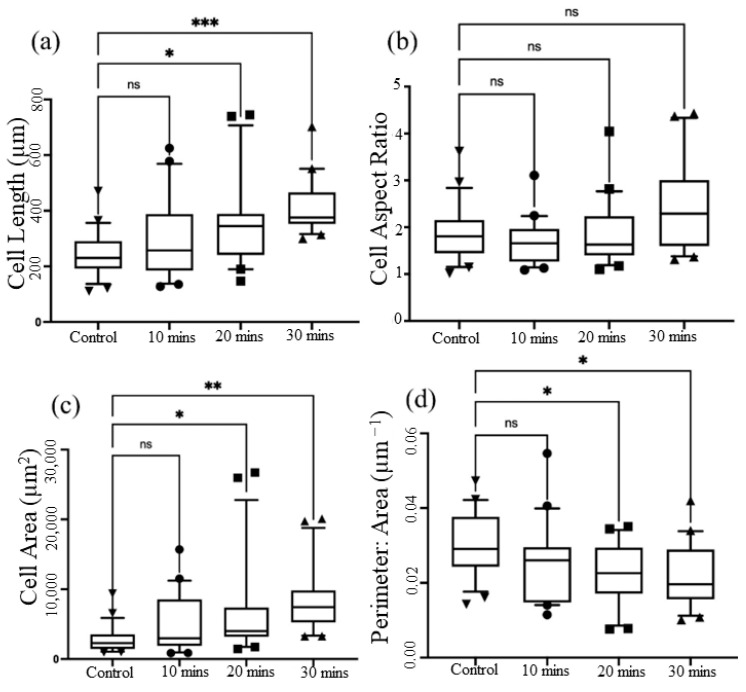
Analysis of key aspects of cell morphology of MDA- MB-231 cells (n = 3): (**a**) cell length; (**b**) cell aspect ratio; (**c**) cell area; and (**d**) perimeter-to-area ratio of MDA-MB-231 cells after stretching for n = 100 cells in each stretching category. The asterisks denote statistically significant differences as determined using multivariate analysis (ANOVA), * (*p* ≤ 0.05), ** (*p* ≤ 0.01), *** (*p* ≤ 0.001) and ns (*p* > 0.05).

## Data Availability

The data presented in this study are available on request from the corresponding author.

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
