# Peer review of "Uniaxial Cyclic Cell Stretching Device for Accelerating Cellular Studies"

_micromachines, 2023, doi:10.3390/mi14081537_

Round 1
Reviewer 1 Report
In this article, the author presented an economical uniaxial tensile equipment and observed the cell shape changes by tensile stress using breast cancer cells. Although he observed various aspects of cell appearance changes due to stimulation, he was unable to discuss essential biochemical parameters closely related to the change or maintenance of cell properties. I believe that in-depth research on the characteristics of stimulated cells is necessary.
1. It is necessary to clarify the goal of this study.
2. When the equipment is put into the incubator, there is a concern about corrosion due to high humidity. I wonder if you have a solution for this.
3. Although PDMS membrane stretches well, it is not an advantageous material for cell attachment. Therefore, it does not match the surface characteristics for smoothing for cell culture that the author talked about in the introduction, and the author's opinion on this matter is needed.
4. In line 323, the author mentioned reaching a maximum in 30 minutes of stretching. However, in figure 5, data for the experimental results after 30 minutes were not presented. To prove the author's point, the experimental data after 30 minutes should be added to figure 5.
5. When the PDMS membrane is pulled in one direction, it is expected that the cells will stretch in a specific direction, as the author experimented, because the membrane stretches in the direction of pulling. However, it is necessary to confirm whether the cells recover to their original state after removing the stretching force or maintain their elongated state.
Author Response
In this article, the author presented an economical uniaxial tensile equipment and observed the cell shape changes by tensile stress using breast cancer cells. Although he observed various aspects of cell appearance changes due to stimulation, he was unable to discuss essential biochemical parameters closely related to the change or maintenance of cell properties. I believe that in-depth research on the characteristics of stimulated cells is necessary.
- It is necessary to clarify the goal of this study.
Reply: The primary objective of the study was to benchmark the new device with the existing one/s. This third-generation device was able to achieve the stretching capability of previous version of stretchers in a short time, hence increasing the efficiency. The following changes have been made in abstract and introduction to clarify the goals.
Abstract, Page 1, Line 20-22 “This third-generation device significantly improved stretching capabilities compared to its previous versions, resulting in a remarkable reduction in stretching time and a substantial increase in overall efficiency.”
Introduction, Page 3, Line 146 -154 “The primary objective of this study is to introduce a cost-effective and high-performance uniaxial cell stretching device, providing a detailed operational concept, experimental design, and characterization. The study focuses on testing the device with MDA-MB-231 breast cancer cells, aiming to demonstrate comparable cellular changes within a 30-minute stretching duration compared to the previously required 2-hour duration. This third-generation device significantly improved stretching capabilities compared to its previous versions, resulting in a remarkable reduction in stretching time and a substantial increase in overall efficiency.”
- When the equipment is put into the incubator, there is a concern about corrosion due to high humidity. I wonder if you have a solution for this.
Reply: High humidity delivers a risk of corrosion to the device. However, all the experiments in this study were conducted in a relatively short timeframe, thus no significant corrosive damage from high humidity was observed. To ensure the device’s longevity in long-term applications, solutions such as applying anti-corrosion paint could be considered.
- Although PDMS membrane stretches well, it is not an advantageous material for cell attachment. Therefore, it does not match the surface characteristics for smoothing for cell culture that the author talked about in the introduction, and the author's opinion on this matter is needed.
Reply: As pointed out by the reviewer, we agree that PDMS may not be the best material for cell attachment. However, we have successfully addressed this limitation by employing a pre-treating method to increase the hydrophilicity of the PDMS membrane, which facilitated cell attachment. PDMS was selected as the material for this study due to its widespread use in fabrication process and well characterizations in cell related research. Additionally, we are actively working on the development of novel materials, specifically porosity-controlled electrospun fiber membranes. These new materials are being explored as potential alternatives to PDMS, aiming to overcome the limitations of PDMS for cell attachment and provide more favourable conditions for cell studies.
- In line 323, the author mentioned reaching a maximum in 30 minutes of stretching. However, in figure 5, data for the experimental results after 30 minutes were not presented. To prove the author's point, the experimental data after 30 minutes should be added to figure 5.
Reply: We made attempts to stretch the cells for a duration of one hour. However, we noted that there was a significant loss in the number of cells during this prolonged stretching period. The data of cell count has been added in the supplementary material Figure S1. In our previous work, (Yadav et al; Experimental Cell Research, Vol 37 doi.org/10.1016/j.yexcr.2019.01.029) we have explained that the cancer cells were reported to undergo apoptosis when they are subjected to compressive stress. Non-cancerous cells, however, tolerate this stress for up to 4 hr of stretching without undergoing apoptosis. The cells not only decrease in count but also loose structural integrity making difficult to calculate the cellular parameters discussed in the figure.
- When the PDMS membrane is pulled in one direction, it is expected that the cells will stretch in a specific direction, as the author experimented, because the membrane stretches in the direction of pulling. However, it is necessary to confirm whether the cells recover to their original state after removing the stretching force or maintain their elongated state.
Reply: The response of the cells to the strain is dependent on the stiffness of the cells. For example, cancers cells have increased rigidity and can undergo apoptosis, however, non-cancerous cells can recover back to their normal stage. We have previously reported this phenomenon and have proposed a method to exploit this phenomenon differentiate between cancerous and non-cancerous cells (Yadav et al, 2019). The response of the cells and recovery to the initial stage is also dependent on the applied strain properties. As mentioned in the introduction (line 59) of this article “For instance, tendon cells anchored to collagen fibres can return to their original size and shape if a small magnitude of tensile stress is applied over an extended period [22,23]. However, these cells may deform permanently or even rupture if they are stretched largely and rapidly.”
Reviewer 2 Report
Micromachines 2023 – Ms 2514771
This manuscript describes a third-generation device tested with a breast cancer cell line to demonstrate cellular changes introduced by stretching forces, permitting that a series of mechanical stimulation parameters be adjusted. The authors propose that their device has the advantage to be positioned on a microscope stage or to be integrated into an incubator during cell culture. It has the potential to process multiple samples simultaneously, thus being of interest for studies of primary cell cultures isolated from clinical samples. The manuscript aligns with the aims of the target journal. Although the Introduction contains a bit long text, it clearly states the research objective and the significance of the study. The other topics are well presented. Minor suggestions: 1. I recommend to introduce “Extracellular matrix” as a keyword; 2. I am not sure whether the identification of the trademarks criticized at lines 71 and 72 should be explicitly mentioned in a scientific paper. Removal of the names of the mentioned platforms would not harm the whole idea of this sentence.
Author Response
This manuscript describes a third-generation device tested with a breast cancer cell line to demonstrate cellular changes introduced by stretching forces, permitting that a series of mechanical stimulation parameters be adjusted. The authors propose that their device has the advantage to be positioned on a microscope stage or to be integrated into an incubator during cell culture. It has the potential to process multiple samples simultaneously, thus being of interest for studies of primary cell cultures isolated from clinical samples. The manuscript aligns with the aims of the target journal. Although the Introduction contains a bit long text, it clearly states the research objective and the significance of the study. The other topics are well presented. Minor suggestions: 1. I recommend to introduce “Extracellular matrix” as a keyword; 2. I am not sure whether the identification of the trademarks criticized at lines 71 and 72 should be explicitly mentioned in a scientific paper. Removal of the names of the mentioned platforms would not harm the whole idea of this sentence.
Reply: Thank you very much. We have made the changes as per the reviewer’s suggestion.
Reply 1: As suggested, we have now added the “Extracellular matrix” as a keyword.
Reply 2: We have removed the name of the trademarks as suggested.
Reviewer 3 Report
There have been developments of mechanical manipulators for cell and tissue biophysical studies, however, the availability of these research tools is still limited, especially in comparison to the broad chemical applications. The authors developed a new generation of cell stretch platform from their previous ones. The uniaxial stretcher has digitalized control at precise strain magnitude and frequency, accessibility to conventional microscopy imaging, and is easily reusable with low cost. Their device is one sound plus to current cell stretch systems.
Here are a few minor revision suggestions:
1) The device is the central piece of this study. The authors have provided many descriptions in text how it works. The reviewer suggests submission of a movie file to record the procedures how the device is assembled and operated during stretching and microscopic imaging.
2) It is unclear whether this platform is able to carry on stretching while the cell samples are imaged simultaneously under (fluorescence) microscope. The authors can provide introduction or discussion to readers, as this function can be exciting to monitor cell biomechanical activities.
3) In Figure 4, the authors want to show cell clustering after stretching. A quantitative measurement can provide more supports to the conclusion. For example, if a cluster is defined as three or more cells are connected, then count the percentage of cells within clusters along the stretching time. The stretch direction is to be added on the images.
4) Line 27: myocytes elongated along the stretch direction. Please double check references as cells are often aligned perpendicular to the stretch direction. (https://doi.org/10.1073/pnas.1221637110; https://doi.org/10.1038/s41598-021-93987-y ).
L178: the medium containing cells…
L265: strain.
The manuscript is well organized structurally and written in English. A few modifications are to be checked as listed above.
Author Response
There have been developments of mechanical manipulators for cell and tissue biophysical studies, however, the availability of these research tools is still limited, especially in comparison to the broad chemical applications. The authors developed a new generation of cell stretch platform from their previous ones. The uniaxial stretcher has digitalized control at precise strain magnitude and frequency, accessibility to conventional microscopy imaging, and is easily reusable with low cost. Their device is one sound plus to current cell stretch systems.
Here are a few minor revision suggestions:
1) The device is the central piece of this study. The authors have provided many descriptions in text how it works. The reviewer suggests submission of a movie file to record the procedures how the device is assembled and operated during stretching and microscopic imaging.
Reply: Thank you for the feedback. As per the reviewer’s suggestion, we have included a supplementary video (S2 in supplementary materials) showcasing the operation of our cell stretching platform. This video provides a more comprehensive visual demonstration of the device in action. Additionally, in Figure 1d, we illustrate how the cell stretching device is positioned beneath a conventional microscope stage, allowing researchers to closely observe and analyse the behaviour of the cells during the stretching process.
2) It is unclear whether this platform is able to carry on stretching while the cell samples are imaged simultaneously under (fluorescence) microscope. The authors can provide introduction or discussion to readers, as this function can be exciting to monitor cell biomechanical activities.
Reply: The device is capable of simultaneous stretching and microscopy. We have shown in figure 1d that the cell stretching device was placed on the stage of the microscope for simultaneous stretching and imaging. Real time monitoring will be possible using environmentally controlled microscope with temperature, CO2 and humidity control for cell cultures. Future application on this area has been now highlighted and added in the conclusion section of the manuscript.
Section 5, Line 376-378 “The device is capable for simultaneous imaging during the stretch cycles providing capability of real time monitoring of biomechanical activities by simply placing the device on the microscope stage.”
3) In Figure 4, the authors want to show cell clustering after stretching. A quantitative measurement can provide more supports to the conclusion. For example, if a cluster is defined as three or more cells are connected, then count the percentage of cells within clusters along the stretching time. The stretch direction is to be added on the images.
Reply: Clustering of the cells has been qualitatively identified not only by visualizing the cell aggregates but also by evaluating the formation of filopodia. We have now quantified the clustering as suggested by the reviewer, however, to mitigate any bias counting two or more connected cells, we counted the number of single cells in a total of hundred cells (originally used to calculate all the cell parameters) and the difference is now reported as percentage of clustered cells. The following information has now been added in the cellular arrangement section.
Section 3.3, Line 317-321 “Proportion of clustering of the cells was calculated subtracting the single cells from total 100 cells counted (used for calculating the cellular parameters). It was noted that only 63% of cells were clustered while stretching for 10 mins, 82% clustering while stretching for 20 mins, 86% clustering while stretching for 30 mins, and 65% of cells were clustered in non-stretched control.”
The stretch direction has been added in figure 4.
4) Line 27: myocytes elongated along the stretch direction. Please double check references as cells are often aligned perpendicular to the stretch direction. (https://doi.org/10.1073/pnas.1221637110; https://doi.org/10.1038/s41598-021-93987-y ).
Reply: The later references provided by the reviewer does mention about perpendicular orientation of the cells to the direction of the stretch. Our results also support the trend (Figure 4). However, the article by Collinsworth AM explicitly mentions that the cells were oriented along the direction of stretch. The most plausible cause of this phenomenon is the cell type, myocytes, however we do not have experience on myocytes and are not able to either prove or disprove the statement on the article referred.
L178: the medium containing cells…
Reply: “the culture media seeded with cells” has been changed to “the culture medium containing cells”
L265: strain.
Reply: “Strain” has been changed to “strain”
Round 2
Reviewer 1 Report
None.